# Deciphering Abnormal Platelet Subpopulations in COVID-19, Sepsis and Systemic Lupus Erythematosus through Machine Learning and Single-Cell Transcriptomics

**DOI:** 10.3390/ijms25115941

**Published:** 2024-05-29

**Authors:** Xinru Qiu, Meera G. Nair, Lukasz Jaroszewski, Adam Godzik

**Affiliations:** Division of Biomedical Sciences, University of California Riverside School of Medicine, Riverside, CA 92521, USA; xinru.qiu@ucr.edu (X.Q.); meera.nair@medsch.ucr.edu (M.G.N.); lukasz.jaroszewski@medsch.ucr.edu (L.J.)

**Keywords:** COVID-19, sepsis, platelets, single-cell RNA-seq, machine learning

## Abstract

This study focuses on understanding the transcriptional heterogeneity of activated platelets and its impact on diseases such as sepsis, COVID-19, and systemic lupus erythematosus (SLE). Recognizing the limited knowledge in this area, our research aims to dissect the complex transcriptional profiles of activated platelets to aid in developing targeted therapies for abnormal and pathogenic platelet subtypes. We analyzed single-cell transcriptional profiles from 47,977 platelets derived from 413 samples of patients with these diseases, utilizing Deep Neural Network (DNN) and eXtreme Gradient Boosting (XGB) to distinguish transcriptomic signatures predictive of fatal or survival outcomes. Our approach included source data annotations and platelet markers, along with SingleR and Seurat for comprehensive profiling. Additionally, we employed Uniform Manifold Approximation and Projection (UMAP) for effective dimensionality reduction and visualization, aiding in the identification of various platelet subtypes and their relation to disease severity and patient outcomes. Our results highlighted distinct platelet subpopulations that correlate with disease severity, revealing that changes in platelet transcription patterns can intensify endotheliopathy, increasing the risk of coagulation in fatal cases. Moreover, these changes may impact lymphocyte function, indicating a more extensive role for platelets in inflammatory and immune responses. This study identifies crucial biomarkers of platelet heterogeneity in serious health conditions, paving the way for innovative therapeutic approaches targeting platelet activation, which could improve patient outcomes in diseases characterized by altered platelet function.

## 1. Introduction

Platelets, also known as thrombocytes, are cell fragments that lack a nucleus. Recent research has shown that platelets play a critical role in the immune system [1,2], acting as first responders during infections and regulating the activation and post-infection deceleration of host immunity. They modulate the function of immune cells through physical adhesion or by releasing cytokines and chemokines, influencing the activation [3,4], proliferation [5], differentiation [6], pathogen clearance [7], and cytokine response of other immune cells [8,9]. Abnormal platelet activity has been implicated in numerous diseases involving immune system dysregulation [10,11,12,13]. This study is a follow-up to our previous work on single-cell transcriptomics in sepsis patients, where we demonstrated that platelets are correlated with patient outcomes [12].

In sepsis, platelets are activated and play key roles in immune response and coagulation. While the overall platelet count drops, the activated fraction of platelets increases in sepsis due to thrombin-mediated activation [14]. The immature platelet fraction (IPF) is also higher in sepsis patients compared to healthy individuals, which could be part of the ongoing inflammatory response to sepsis [15,16].

In severe COVID-19, similar to sepsis, there is a trend of decreased platelet counts, particularly in patients with severe disease [17]. A meta-analysis of 7613 COVID-19 patients revealed that patients with severe disease had a lower platelet count than those with non-severe disease. This decrease in platelet count could suggest a worsening thrombotic state, with lower nadir platelet counts associated with increased mortality [18]. At the same time, there is evidence of increased platelet activation in COVID-19. Platelets are hyperactivated in COVID-19; however, the mechanisms promoting this activation are not fully understood [19]. Damaged lung tissues and pulmonary endothelial cells may activate platelets in the lungs, resulting in aggregation and the formation of microthrombi [20]. Furthermore, there is growing evidence of platelet hyperactivation contributing to the severity of COVID-19. Increased levels of platelet and platelet-leukocyte aggregates have been linked to COVID-19 severity.

Platelets are activated in patients with SLE and contribute to the immune response and inflammation [13]. Increased platelet activation and platelet-complement interactions have been associated with vascular events, both venous and arterial, in SLE patients [21]. Platelets in systemic lupus erythematosus (SLE) can respond to immune complexes, complements, and damage-associated molecular patterns, and they are an important source of biologically relevant molecules in the circulation [13].

Single-cell RNA sequencing (scRNA-seq) has been used to study the transcriptome of isolated peripheral blood platelets from patients with various diseases, including periodontitis and diabetes [22], ovarian cancer [23], and COVID-19 [24]. Utilizing single-cell analysis, we aimed to identify and characterize subpopulations of abnormal platelets, investigate the molecular footprint of their pathogenic changes, and develop testable hypotheses regarding their contribution to patient outcomes, including fatality or survival. Our research findings could pave the way for targeted therapies that specifically address these abnormal platelets, potentially improving clinical outcomes for affected patients.

## 2. Results

### 2.1. PBMC Composition Changes with Patient Disease Severity and Outcome

First, we analyzed the relative populations (fractions) of immune cell subsets in peripheral blood mononuclear cells (PBMC) in relation to disease severity, outcome, and disease type (Figure 1A). The population proportions of platelets, precursor cells, and erythroblasts in PBMCs increased with the disease and outcome severity (Figure 1B, Appendix A), while the T cell fraction decreased (Figure 1C). B cell fractions were significantly increased in SLE patients, which is a known phenotype in SLE patients (Figure 1D); monocyte and neutrophil fractions were significantly increased in convalescence patients (Figure 1E,F); however, fractions of the dendritic cells decreased as outcome severity increased (Figure 1G). We evaluated ratios of different cell types as a criterion for separating fatal (FT) and survivor (S) patients and found that the ratio of platelets to T cells, (Pla-T ratio) had the highest area under the curve (AUC) at 0.754, with a 0.063 ratio between FT and S patients (Figure 1H).

### 2.2. XGBoost and Deep Neural Network Modeling Identifies Biomarkers of Survival and Fatal Platelets

In our study, we employed two machine learning approaches—a Deep Neural Network (DNN) and an eXtreme Gradient Boosting (XGB) algorithm—to analyze and interpret with the objective of identifying key biomarkers. These biomarkers are indicative of patient survival and the presence of fatal platelets, which are critical in the prognosis and treatment of life-threatening conditions. When comparing the two models, the XGB model has a slightly higher accuracy, while the DNN model shows marginally better performance in F1 score, precision, and recall. This could suggest that while XGB is slightly better at correctly identifying both positive and negative classes, DNN is better at identifying positive cases when the outcome is positive and also better at not missing positive cases (Figure 2A). We only obtained the features that are within the top 5% of importance or gain from their respective models. We also looked into genes that had an absolute log2fc > 1. The overlap of 21 biomarkers suggests that there is a consensus between the three models on these biomarkers’ significance in relation to the outcome of interest (e.g., survival or fatal platelets) (Figure 2B). In order to discern the specific features that could potentially serve as biomarkers indicative of either survival or fatal outcomes, we labeled each gene within a volcano plot, as presented in Figure 2C of our study. The biomarkers identified in survival platelets include *AIF1, FOS, CD74, JUN, JUNB, HLA-DRA, MNDA, RPL39, RPS21, RPS18, EEF1A1, RPS28, RPL34, S100A8, S100A11*, and *S100A12*. The biomarkers identified in fatal platelets include *HBA2, HBB, HPSE SLC25A37,* and *TMCC2*. We conducted a gene ontology enrichment analysis on the identified genes. For genes associated with survival, the enrichment analysis revealed a strong connection to processes involved in cytoplasmic translation and the active immune response, as detailed in Figure 2D. In contrast, genes related to fatal outcomes were predominantly associated with pathways involved in coagulation and reactive oxygen species (ROS) metabolic pathways, as depicted in Figure 2E.

### 2.3. Platelets Amplify Endotheliopathy and Disseminated Intravascular Coagulation in Fatal Patients

Platelets have been shown to play a role in the development and progression of endotheliopathy and disseminated intravascular coagulation (DIC) [25,26]. Platelets can bind to and activate endothelial cells, releasing pro-inflammatory and pro-coagulant molecules that can contribute to the development of DIC. Additionally, platelets can also contribute to thrombus formation and further damage to the endothelium [27].

We looked into the expression of the Integrin Subunit Alpha 2b (*ITGA2B*) gene, which is involved in platelet-endothelial cell interactions by binding to the Integrin Subunit Alpha V (*ITGAV*) [28]. The patients who ultimately passed away had the highest levels of *ITGA2B* expression, followed by the patients who were in a severe disease state (Figure 3A). On the pathway level, the reorganization of the actin cytoskeleton (GO:0007010) also followed the same severity trend (Figure 3B). DIC is a complex condition characterized by abnormal clotting and bleeding due to the activation of the coagulation cascade and the depletion of clotting factors and platelets [29]. Several other GO terms associated with DIC, such as blood coagulation (GO:0007596), inflammatory response (GO:0006954), extracellular matrix disassembly (GO:0022617), and platelet activation (GO:0030168), had the highest expression in FT/SV patients. All modules had lower expression in CV and SLE patients (Figure 3C).

### 2.4. Platelet Subpopulations Associated with Disease Severity

The standard analysis of scRNA-seq focuses on identifying clusters that overlap with the standard cell types. Here we carried the clustering further to understand the details of platelet population changes in diseases. By trial and error, we identified similarity thresholds that resulted in stable clusters with the best separation and the minimal signature pathway overlap, resulting in thirteen clusters, designated C0 through C12 (Figure 4A,B). We computed the composition of each cluster in terms of the contributions from different outcome groups. The C11 cluster was strongly associated with the FT group, with 78% of all C11 cells originating from patients with fatal outcomes. Clusters C3, C5, and C9 had the highest proportion of cells from healthy controls (17%), while clusters C6 and C11 had the lowest contribution (1%) from HC. Cluster C3 had the highest proportion of cells from the mild disease group at 25%. In clusters C6, C7, C10, and C12, the severe group comprised over fifty percent of total platelets (Appendix A). Over 70% of platelets in clusters C6, C8, and C10 come from the samples from the survivors (Appendix A). Based on these results, we designate clusters C4, C9, and C11 as “fatal”, C8 as “convalescent”, and C6 and C10 as “survival” (Figure 4C,D).

### 2.5. Characteristics of Platelet Subpopulations

We used DEG and Gene Set Enrichment Analysis (GSEA) to identify molecular factors and pathways that could help us differentiate between the clusters (subpopulations) of platelets identified in the previous step. The sets of genes enriched in the fatal platelet clusters C4 and C11 were significantly different from those in other platelet clusters (Figure 5A), and in some cases, they showed opposite trends from those in the convalescent and survival clusters. The fatal cluster C4, whose signature module comprises *HPSE*, *WFDC1*, and *PF4* genes, and the convalescence cluster C8, whose signature module consists of *RPLP0, RPS6*, and *RPS23*, have several pathways that trend in the opposite direction. Fatal cluster C11, whose signature module includes *TPT1, POLR2L*, and *CSRP1*, had the highest energy consumption pathway scores, including oxidative phosphorylation (OXPHOS) and glycolysis (Appendix A), all the same, the lowest inflammatory response score (Appendix A). The other fatal cluster C9, whose signature modules include *HBB, HBA2,* and *HBA1*, showed the weakest interferon response, including alpha and gamma interferons (Appendix A). Coagulation, epithelial-mesenchymal transition (EMT), and the apical junction are all at their highest levels in C4 but at their lowest levels in C8 (Figure 5B–D). At the same time, MYC targets v1 and v2, which contain nuclear-encoded genes involved in mitochondrial biogenesis [41], are the lowest in C4 yet, highest in C8 (Appendix A). Based on these findings, we can conclude that platelets from the fatal cluster C4 are highly active in angiogenesis, coagulation, and endotheliopathy, while having the lowest RNA processing, cell division, and mitochondrial biogenesis, while those from the convalescence cluster C8 had the opposite trend in all these pathways.

Cluster C6, associated with survival, was characterized by elevated expression of genes *CRBN*, *CD58*, and *SRSF3* and exhibited the most significant activity in the Notch signaling pathway alongside minimal *IL6/JAK/STAT3* signaling pathway activity, as shown in Appendix A. Conversely, survival cluster C10, distinguished by a gene module including *CD74*, *HLA-DRA*, and *CD79A* related to antigen presentation, demonstrated higher activity in allograft rejection and MYC targets v2 pathways (Appendix A). Both clusters C6 and C10 showed reduced activity in angiogenesis, coagulation, epithelial-mesenchymal transition (EMT), and apical junction pathways, which were notably upregulated in the fatal cluster C4, as detailed in Figure 5B–D.

Then, we examined the GO terms up-regulated in the fatal clusters C4, C9, and C11. The key pathways up-regulated in fatal cluster C4 include wound healing, platelet activation, hemostasis, and coagulation, which is consistent with the known observations that thrombotic problems are a major cause of morbidity and mortality in COVID-19 patients [42]. Oxygen transport, hydrogen peroxide metabolism, gas transport, and erythrocyte development are enriched terms for C9, reflecting the hypoxic environment of C9 platelets (Figure 5F). ATP metabolic activities such as oxidative phosphorylation, cellular respiration, and aerobic respiration are among the enriched terms in C11; this is consistent with the C11 GSEA results, which suggested that C11 platelets are inactive (Appendix A). Platelets in quiescence are known to require ATP for their basic function. According to a study, glycolysis produces up to 65% of the required ATP in inactivated platelets, with mitochondria providing the remainder [43]. C4 is the most active cluster in FT groups. Therefore, we examined C4-enriched genes associated with diseases such as arteriosclerosis, bacterial endocarditis, thrombosis, and frontotemporal lobar degeneration, a neurodegenerative disorder (Figure 5G). Among the mechanisms shared by survivor clusters C6, C8, and C10 are platelet translation regulation, mRNA processing, and ribosomal RNA biogenesis (Figure 5H, Appendix A). Additionally, C10 was enriched for the GO terms homeostasis, lymphocyte activation cells, and leukocyte antigen cell-cell adhesion (Appendix A).

### 2.6. Pseudotime Trajectory Analysis Identifies Platelet Signature Dynamics in Survival or Fatal Disease Outcomes

The fatal cluster C4 appears to be critical for the negative outcome groups, so we evaluated possible events leading to its emergence. Pseudotime analysis, although unable to recover real time dynamics of cell populations, provides hints as to their order along the developmental trajectories [42]. The possible precursor of C4 is the C0 cluster, which is also close to the C1 and C6 clusters in the trajectory map (Appendix A). The C0 cluster has two possible developmental routes; fatal C4 or survival C1 and C6 (Figure 6A,B), indicating that the platelets in the C0 could be targeted by early intervention to inhibit their development into the C6 phenotype.

Then, we examined DEGs between C4 and C0 and C0 and C1. *AKR7A2, CALD1, CALR, CD36, CSRP1, CYBA, ENSA, FCGR2A, HBG2, HCST, HMGN1, HPSE, MMRN1, NDUFA4, PF4, RPLP1, RPS9, SAMD14,* and *TPT1* genes are consistently up-regulated, with a log2Fold change greater than 0.4 and an adjusted *p*-value less than 0.05, from C0 to C4, and C1 to C0 (Figure 6C,D). For further analysis, we defined a fatal platelet module consisting of these genes. Among the fatal module-enriched GO terms are regulation of body fluid levels, coagulation, phagocytosis, and tumor necrosis factor (TNF) production (Appendix A). The DEGs consistently down-regulated between C4 vs. C0 and C0 vs. C1 were *ADIPOR1, CDKN1A, MAP3K7CL, MMD, NEAT1, NUTF2, PTMA, RAB31, SLC40A1, TMEM140,* and *TSC22D3*. They were used to define the platelet survival module. However, no GO, KEGG, or Reactome pathways were enriched in the survival module. The scores for the fatal and survival modules were then calculated for all disease clusters and disease severity levels. As expected, cluster C2 had the lowest fatal score, followed by C1 and then survivor cluster C6. C11 had the greatest fatal score, followed by C4 and C9 (Figure 6E). C2 had the greatest survival module score, followed by C1 and survivor cluster C6, while C11 and C4 had the lowest scores for the fatal modules (Figure 6F).

The fatal and survival module scores could also serve as indicators of the disease outcome. The patient group with the highest fatal scores was the fatal group, followed by the severe group, while the mild group received the lowest score on the fatal module, followed by the moderate group. The HC group was positioned in the center of the fatal module (Figure 6G). The groups with the lowest survival module ratings were fatal and severe groups. However, the highest scores were in the SLE and mild groups (Figure 6H). When platelets encounter an immunological disorder response, their expression shifts away from HC, as demonstrated by the findings. In addition, we investigated the composition of platelet subclusters and discovered that the fraction of fatal cluster C4 was the best indicator among clusters for distinguishing between S and FT patients (Figure 6I), with an AUC of 0.749. When C4 platelets exceeded 3.36 percent of the PBMC platelet composition, patients were at risk of death. In this case, detecting the presence of platelet C4 in a timely manner could help in the patient’s survival.

These findings show the complexity of COVID-19 and sepsis in relation to gene and pathway signatures in platelets, especially for patients with severe active disease as well as those recovering from it. This research could lead to a better understanding of platelet processes in COVID-19 and sepsis patients, which could help guide therapeutic options for both patient subgroups.

### 2.7. Unique and Shared Gene Expression Changes in Platelets from COVID-19, SSH, Sepsis, and SLE Samples

Comparing COVID-19, SSH, sepsis, and SLE platelets to HC platelets, we identified differentially expressed genes (DEGs) and related pathways (whose up- and down-regulations were analyzed using the GO and KEGG databases). There were 45 DEGs that were up-regulated in all four diseases studied here, including *IFI27L2, IFITM2, IFITM3*, and S100 family genes (*S100A8* and *S100A9*) and genes for ATP synthase, such as ATP5E and *C9orf16. BEX3, LCN2, RHEB*, and *TMEM219* are genes associated with apoptosis (Appendix A) (Appendix A). 271 DEGs were downregulated in COVID-19, 515 DEGs were downregulated in SSH, 901 DEGs were downregulated in Sepsis, and 747 DEGs were downregulated in SLE. 129 genes were downregulated in all four diseases (Appendix A). The 129 genes included 68 ribosome-related genes. *CD52* is a therapeutic target and predictive biomarker for sepsis (14). *CD3E, CD48, CD7, LCK, LEF1, PTPRC*, and *TCF7* are required for the activation of T cells (Appendix A).

There were 294 pathways up-regulated in COVID-19, 346 in SSH, 394 in sepsis, and 139 in SLE as compared to HC (Figure 7A). There were 47 pathways that were consistently up-regulated in the four considered diseases. There were 120 down-regulated pathways in COVID-19, 381 in SSH, 710 in sepsis, and 657 in SLE (Figure 7B). Among them, 81 pathways were consistently down-regulated by all four considered diseases.

To better interpret the functional meaning of the consistently up/down-regulated pathways among the four diseases, we used R package rrvgo, which is heavily influenced by REVIGO [42] to summarize the 47 up-regulated pathways and 81 down-regulated pathways from COVID-19 vs. HC, SSH vs. HC, Sepsis vs. HC, and SLE vs. HC. Pathways up-regulated in platelets consistently among the four considered diseases include GO: neutrophil-mediated immunity, KEGG: Parkinson disease, and GO: ATP metabolic process (Figure 7D). All translation processes are down-regulated, including the GO: nuclear-transcribed mRNA catabolic process, nonsense-mediated decay, GO: regulation of translational initiation, GO: protein localization in the endoplasmic reticulum, KEGG: ribosome, and GO: response to interleukin-4 (Figure 7D). Interleukin-4 has several biological activities, including the stimulation of activated B cells and T cell proliferation, mediated by pathways such as T cell activation regulation and response [43].

We then focused on the pathways that were up-regulated in both COVID-19 and sepsis relative to HC. KEGG: Parkinson disease, KEGG: Huntington disease, KEGG: Prion disease, KEGG: Alzheimer disease, KEGG: pathways of neurodegeneration, multiple diseases, and KEGG: Amyotrophic lateral sclerosis were consistently up-regulated in platelets from COVID-19 and sepsis (Appendix A). COVID-19 has been recently associated with neurological diseases [45], even though the molecular mechanism of this association is not clear. As with sepsis, COVID-19 may act as a significant inflammatory insult, increasing the brain’s susceptibility to neurodegenerative illnesses, cognitive decline, and the likelihood of acquiring dementia later in life [46]. However, up-regulation of the pathways seen in neurons in neurogenerative diseases in platelets may suggest an effect of common regulatory mechanisms rather than a direct effect of abnormal platelets on the brain.

The KEGG pathways related to endocytosis and Fc gamma-R-mediated phagocytosis were found to be up-regulated, indicating an active engagement of platelets in the internalization of virions. This internalization is facilitated by Toll-like receptors binding to virion-released lysosomal ligands such as single-stranded RNA, double-stranded RNA, and CpG DNA, which trigger platelet activation and the release of granules, leading to the exposure of P-selectin and the formation of platelet-leukocyte aggregates. Additionally, pathways linked to protein targeting, translation, and ribosomes were identified, alongside a noted down-regulation in the context of sepsis for KEGG: Coronavirus disease—COVID-19. This down-regulation is primarily attributed to the diminished expression of ribosomal proteins, a pattern observed in both COVID-19 and sepsis conditions, as elaborated in Appendix A.

Then, we investigated the pathways that are enriched in COVID-19 and sepsis fatal patients (FT) in comparison to survivors (S) (Figure 7C). GO: neutrophil-mediated immunity and GO: ATP metabolic process were up-regulated in FT patients and were likewise up-regulated in disease vs. HC pathways. GO: response to endoplasmic reticulum stress; GO: response to hypoxia; and GO: intrinsic apoptotic signaling pathway in response to oxidative stress are up-regulated stress pathways. While the KEGG: Bacterial invasion of epithelial cells was also among the up-regulated pathways, we concluded that only the part of this pathway involved in cytoskeleton rearrangement is upregulated (see discussion). GO: antigen processing and presentation of peptide antigen via MHC Class I were also up-regulated in fatal patients with COVID-19 and sepsis. Platelet MHC Class I mediates CD8+ T-cell suppression during sepsis, according to a previous study [47]. Pathways such as GO: protein localization to the endoplasmic reticulum, GO: regulation of translational start, and GO: regulation of RNA stability were down-regulated. In addition, lymphocyte activation pathways, such as GO: regulation of T cell proliferation, GO: interferon-gamma-mediated signaling pathway, and GO: regulation of interleukin-12 production, were down-regulated in FT patients. Interferon-gamma is mostly secreted by activated lymphocytes, including CD4 T helper type 1 cells and CD8 cytotoxic T cells [48], whereas interleukin-12 is known as a T cell-stimulating factor [49] (Figure 7C). These data indicate that platelets from the disease cohorts exhibited less immunological activation, fewer translational activities, and more neurodegenerative tendencies, such as the KEGG enrichment for Parkinson’s disease (Figure 7D). Compared to survivor platelets, the aforementioned tendencies became more pronounced in FT platelets, which began to exhibit the ability to invade epithelial cells (Figure 7C).

### 2.8. Pathway Enrichment Related to Disease Severity in Platelets

Given the dynamic gene expression changes in platelets in multiple diseases, we evaluate what gene expression modules were significantly changed depending on disease severity. Two modules that diminish with disease severity are MHC class II genes and translation initiation (Table 1). MHC Class II scores are much higher in convalescent patients than in healthy controls, indicating that MHC Class II can be utilized as an indicator of patient recovery. High case fatality rates of COVID-19 reported in some countries have been linked to inadequate MHC class II presentation and, consequently, a weak adaptive immune response against these viral envelope proteins, according to studies [50]. The lowest score for translation initiation modules (GO:0006413) was found in the platelets of patients who did not survive the diseases, indicating a halt in protein translation due to fatal illness. An interesting observation is the presence of pathways implicated in neurodegeneration (GO:0070843) in severely sick COVID-19 patients (Appendix A). Except for axonal transport modules, platelets from severe and fatal COVID-19 patients show all the trends observed in neurodegenerative diseases’ major biological processes. In both sepsis and COVID-19 patients, neurodegeneration-related pathways became more severe as the disorders advanced. The scores for blood coagulation (GO:0007596), platelet activation (GO:0030168) (Figure 3C), oxidative phosphorylation (OXPHOS), and glycolysis (Table 1) (35) modules confirmed the hypothesis that platelet coagulation and energy consumption are functionally linked to the severity of sepsis disease and the progression of COVID-19 disease [51,52]. Moreover, platelets in convalescent patients had higher glycolysis scores, which corresponded to module scores in response to oxygen radicals (Appendix A), indicating that platelets in convalescent patients were also hypoxic. Hypoxia induces oxidative damage to neural cells and causes widespread neurodegeneration [53].

Interferon response modules, such as response to type I IFN (GO:0034340), IFN-β (GO:0035456), and IFN-γ (GO:0003341) (Table 1), had the highest scores in moderate patients and the lowest scores in HC. The interferon (IFN) protein family is crucial for the immune response against viruses and other infections. IFNs have been demonstrated to play a significant role in preventing SARS-CoV-2 infection in the context of COVID-19, but they have also been linked to severe symptoms [54]. The findings may explain the contradictory reports of COVID-19 patients with impaired and robust type I IFN responses. Although robust type I IFN responses have been reported in patients with severe COVID-19 [55], studies have demonstrated that type I IFN responses are severely impaired in the peripheral blood of patients with severe or critical COVID-19, as indicated by low levels of type I IFNs and interferon-stimulated genes [56]. Our data indicate that severe and fatal patients had IFN levels much lower than moderate patients but greater than mild patients and healthy controls. In contrast to individuals with other disorders, COVID-19 patients have continuously elevated IFN levels. The above data conclude that the higher expression of MHC Class II and translational initiation expression in platelets means better outcomes in patients. Coagulation and higher ATP synthesis from platelets mean worse outcomes for patients. As for the interferon response, with both protective and deleterious effects being reported, we confirmed the theory that severe COVID-19 is associated with decreased IFN signaling [57,58].

### 2.9. The Platelets’ Crosstalk with Monocytes and Lymphocytes

An important role of platelets is communication with other cell types during the formation of the thrombi, both regular ones forming during hemostasis and potential abnormal ones forming during sepsis and COVID-19. Using the ligand and receptor database from iTalk [59], we evaluated these interactions by computing the product of average ligand and receptor expressions in the corresponding cell types from peripheral blood mononuclear cells (PBMC). The platelet-monocyte interaction was evaluated and found to have the highest score in fatal patients relative to other outcomes (Figure 8A). Consistent with our previous sepsis study [12], platelet-monocyte interaction scores were significantly elevated in FT sepsis patients. This phenomenon was also noticed in COVID-19-severe patients. Compared to control participants and mildly infected individuals, ICU-admitted COVID-19 patients had higher platelet-monocyte aggregate levels [52]. Thus, we postulate that the aggregation of platelets and monocytes is linked with the severity and mortality of sepsis and COVID-19.

Chemokine receptor 2 (CCR2), CCR5, and their selective ligands, chemokine ligand 2 (CCL2), and CCL3, have been found to promote the trafficking of leukocytes to sites of inflammation and regulate their activation [58]. Also, the CCL2-CCR2 and CCL3-CCR5 ligand-receptor systems in differentiating T cells have been identified [60]. CCL2-CCR2 and CCL3-CCR5 interactions between platelets and T cells revealed that SLE had the highest expression of the CCL3-CCR5 system between platelets and T cells, a characteristic of SLE [61]. Compared to other outcome groups, the CCL3-CCR5 system’s expression was the lowest among patients who did not survive the diseases (Figure 8B). The results are associated with T cell differentiation module scores (GO:0030217) (Figure 8F). The CCL2-CCR2 and CCL3-CCR5 ligand-receptor systems are more prevalent in surviving patients than in HC (Figure 8C).

CD40 is recognized to play a crucial role in B lymphocyte proliferation and differentiation [62]. CD40 ligand CD40LG expression is low or undetectable on the surface of resting platelets but is highly expressed upon platelet activation [63]. Also, platelets are noted to directly influence adaptive immune responses via the secretion of CD40 and CD40L molecules [2]. Platelets and B cell interaction analysis revealed that healthy controls had the highest CD40 induction relative to other groups, whereas patients who did not survive had the lowest CD40 induction from platelets and B cell interaction (Figure 8E). The module score for B cell proliferation (GO:0042100) (Figure 8G) also supported this conclusion. SLE also had the highest expression of CD40LG and CD40 interaction between platelets and B cells compared to other diseases (Figure 8D). These data implicate that in SLE, platelets may induce B cell activity, consistent with previous studies reporting the putative effects of activated platelets in SLE pathogenesis [13].

## 3. Discussion

For this study, we compiled single-cell transcriptome datasets from multiple cohorts of patients with inflammatory diseases with infectious and non-infectious roots. Extensive evidence points to a key role for platelets in all these diseases, severe COVID-19 [10,64], fatal sepsis [12], and SLE [13]. In our analysis, we identified both shared and unique signatures in platelet transcriptomics in patients with these diseases, defining unique platelet subsets that correlate with disease outcomes. The ratio of platelets to T cells (Pla-T ratio) was shown to be a useful criterion for separating patients with fatal and survival outcomes (Figure 1H). Our study utilized DNN and XGB machine learning models to pinpoint biomarkers correlating with patient survival and the occurrence of fatal platelets, finding consensus on 21 key biomarkers, with XGB slightly outperforming in accuracy and DNN in F1 score, precision, and recall. Gene ontology enrichment revealed that survival-linked genes are involved in cytoplasmic translation and immune response, while fatal-linked genes are associated with hypoxia and coagulation processes (Figure 2).

Analysis of the upregulated pathways shows that pathways characteristic of the development and progression of endotheliopathy and disseminated intravascular coagulation (DIC) are upregulated in platelets. The patients who passed away had the highest expression of the genes involved in platelet-endothelial cell interactions and the highest expression in the modules related to DIC, including actin cytoskeleton reorganization (Figure 3). Platelet actin cytoskeleton reorganization plays a role in this process, as the cytoskeleton helps to determine the shape and function of platelets and is essential for platelet activation and aggregation, and changes observed here contribute to the pathological variants of this process.

We identified subgroups of platelets overrepresented in convalescent, surviving, and fatal patients. Specifically, the three types of platelets were highly represented in FT patients—the coagulation cluster C4, hypoxic cluster C9, and quiescence cluster C11. This suggests that a combination of anticoagulants, anti-hypoxic therapies, arteriosclerosis treatments, and selective serotonin reuptake inhibitors (*SSRI*) should be considered for the removal of fatal platelets. We examined the pseudotime trajectory of platelet clusters and identified C0 as the cluster preceding fatal cluster C4. C0 can be differentiated by following the path leading to the emergence of either a fatal (C4) or survival (C1 leading to C6) cluster. We propose that targeting C0 and modifying its developmental path may prove to be an effective strategy to treat immune instability in sepsis or COVID-19. Expression of *MYL9, FCER1G,* and *PARVB* are the signatures of cluster C0 (Figure 4E) (Appendix A). *MYL9* was investigated in relation to platelet dysfunction [65]. Comparing the fatal cluster C4, its ancestor C0, and its alternative, survival cluster C1, to each other, we identified two gene modules associated with fatal outcome and survival and labeled them accordingly (Figure 6C,D). Some genes in the fatal module, such as heparanase (*HPSE*), had already been studied as targets of therapy aimed at selectively neutralizing platelets to mitigate disease severity. HPSE expression and activity were shown to increase in platelets during clinical sepsis at both the transcriptomic and proteomic levels [66]. The same phenomenon was also observed in COVID-19 patients’ plasma [67]. Thus, in the fatal and survival module genes identified in this study, there are potential clinical therapeutic targets and diagnostic and prognostic biomarkers in platelets that are worth investigating for future studies.

The study also compared gene expression in three diseases (COVID-19, Sepsis and SLE), SSH, and healthy controls and found consistent changes in gene expression, including stress pathways (Figure 7D) and decreased levels of MHC class II gene expression in fatal cases (Table 1). The presence of pathways also seen in neurodegenerative processes in the brain is also upregulated in severe COVID-19 patients and in all immune response disorders. Similarly, the ATP metabolic process is up-regulated, but ribosome biogenesis and lymphocyte activation are commonly down-regulated (Figure 7C).

Platelet-monocyte aggregation has been studied as a hallmark of severe COVID-19 [64,68,69] and sepsis [12,70,71]. The findings agree with our cell-cell interaction study (Figure 8A), which again provides the molecular mechanism of this effect. In addition, we discovered that platelets could influence lymphocyte activation, proliferation, and differentiation through interactions with other immune cells, implying that platelets can modulate lymphocyte function and contribute to inflammatory and immune responses.

Overall, the study highlights the role of platelets in the development and progression of disease and the importance of monitoring changes in immune cell subsets and gene expression for predicting disease outcome and severity. We were able to study the etiology of dysfunctional platelets in sepsis, COVID-19, and SLE patients due to the high number of clinical samples included in our integrated analysis. Our research demonstrates the significance of platelets’ dysfunction in immunological imbalance disorders and supports the utility of platelet-directed therapies to treat multiple immune disorders.

## 4. Materials and Methods

### 4.1. Integrated Single-Cell Transcriptome Atlas of Peripheral Blood Mononuclear Cells (PBMCs) from COVID-19, Sepsis, and Systemic Lupus Erythematosus Patients

We gathered single-cell RNA-seq datasets of peripheral blood mononuclear cells (PBMCs) from COVID-19 [30,31,32,33,34,35,36,37,38], sepsis [12,39], and systemic lupus erythematosus (SLE) [40] patients, for a total of 413 samples, in order to study and compare the immunological dysregulation caused by these diseases. The COVID-19 datasets sourced include GSE150728, GSE155673, GSE158055, GSE151263, and E-MTAB-10026, which were acquired from the GEO database and ArrayExpress. Blood PBMC datasets from COVID-19 patients were also downloaded. For the sepsis studies, datasets GSE163668 and GSE167363 were obtained from the GEO database. The systemic lupus erythematosus (SLE) study involved dataset GSE142016, also downloaded from the GEO database.

In the COVID-19 studies, individuals with severe influenza, lung infections testing negative for COVID-19 were identified as the “hospitalized patients with similar symptoms” (SSH) group, and we included them in our analysis. We categorized the samples into six groups based on the severity of the disease: healthy control (HC), convalescence (CV), mild (ML), moderate (MD), severe (SV), and fatal (FT), and created a separate group for systemic lupus erythematosus (SLE). For the outcome analyses, we grouped patients who survived from the CV, ML, MD, and SV groups into a survivor (S) group, with the patients with unannotated outcomes forming another group labeled as the unknown group (Figure 1A, Table 2).

### 4.2. Removal of Doublets in scRNA-Seq Data and Identifying the Cell Types

While platelets can form aggregates with monocytes within peripheral blood mononuclear cells (PBMCs), particularly in states of inflammation, the platelets used in our study were isolated as single cells. To reduce the events when two or more cells are incorrectly classified as a single entity, we excluded cells exhibiting more than 6,000 unique genes expressed [72]. Furthermore, we ensured the accuracy of platelet identification by using both source data annotation and the platelet cell marker pro-platelet basic protein (*PPBP*). To enhance our analysis, we employed the bioinformatics tools Seurat (v.4.3.0) [73] and SingleR (v.1.10.0) [74]. Seurat (v.4.3.0), a powerful tool for single-cell RNA sequencing data analysis, facilitated cell type classification by projecting reference metadata onto a query object without altering the query expression data. After identifying the anchor points, we used Seurat’s TransferData() function, which provided us with a matrix of predicted IDs and prediction scores. This process allowed us to categorize our query cells based on the reference data. SingleR (v.1.10.0) was another integral part of our toolkit. This tool functions by comparing the expression profile of each single cell in the query dataset with each of the reference expression profiles. By calculating the similarity between the query cell and each reference cell, SingleR (v.1.10.0) identifies the most likely cell type for each single cell in the query dataset. This comparison allows for precise and granular cell type annotation, which is essential for the detailed single-cell analyses conducted in this study.

### 4.3. Integrating the Datasets

Expression data for platelets was extracted from all the datasets identified in the previous step. Using HGNChelper (v.0.8.1) [75], we harmonized the nomenclature of the genes from different datasets and identified the 3000 most variable genes from each. For the integrated analysis, the top 3000 highly variable genes were retained. Harmony (v. 0.1.1) [76] was executed with PCA embeddings (30 PCs) as input and the default parameters to eliminate batch effects among twelve datasets (nine COVID-19 datasets, two Sepsis datasets, and one SLE dataset). Seurat was then applied to the Harmony embeddings to determine the clusters. In the round of clustering, the resolution for Louvain clustering was set to 0.4.

### 4.4. Differential Expression Analysis

The MAST (v.1.22.0) method [77] from the Seurat package (v.4.3.0) (implemented in the FindAllMarkers function) was used with the default parameters to perform the differential gene expression analysis and identify the DEGs. We used patients’ categories, as described before, to compare DEGs between disease severity states.

### 4.5. Machine Learning Models

eXtreme Gradient Boosting (XGB) [78] is an advanced decision-tree-based algorithm that enhances the traditional decision tree model by building a series of successive trees where each new tree corrects the errors made by the previous ones, reducing prediction errors through a technique known as gradient boosting. XGB’s architecture allows for the incorporation of new input data, in this case, gene expression data from platelets classified according to survival or fatal outcomes, to iteratively refine the model. XGB’s inherent feature selection capability, attributed to its bagging property, enables us to use all gene expression values to automatically determine the most informative features. To optimize the model’s parameters, a grid search was employed to explore a wide array of hyperparameter combinations, such as learning rate, number of trees (estimators), and tree complexity (maximum depth), settling on a learning rate of 0.1 and a maximum depth of 9 for our model. The model’s predictive power was evaluated using a confusion matrix. The XGBoost model was developed within the R environment, utilizing the XGBoost package (v. 1.7.5.1).

In parallel, a Deep Neural Network (DNN) model was developed using the Keras and TensorFlow Deep Learning frameworks. The model’s hyperparameters, including the number of epochs, batch size, choice of optimizer, weight initializer, quantity of hidden layers, and number of nodes per hidden layer, were tuned through a comprehensive grid search approach. This method tested numerous hyperparameter combinations to determine those that yielded the highest accuracy. The significance of each feature within the DNN model was assessed based on its impact on model accuracy when its values were permuted; a more substantial decline in accuracy indicates a higher feature importance. This DNN model implementation was performed in R, leveraging the capabilities of the ‘caret’ package (v. 6.0-94).

### 4.6. Comparison of Module Scores

We measured the extent to which individual cells expressed certain predefined expression gene sets using cell module scores. All calculations and comparisons of module scores were performed using the AddModuleScore function from the Seurat package with the default parameters. We compared the expression of modules extracted from the GO.db database [79]. Additionally, we used literature-based modules for oxidative phosphorylation (OXPHOS), glycolysis [80], and MHC class II. We disregarded genes without detectable expression in our data.

### 4.7. Pathway Enrichment Analysis

We analyzed Gene Ontology biological process terms (GO-BP) and Kyoto Encyclopedia of Genes and Genomes (KEGG) pathways using R package clusterProfiler (v.4.4.1) [81] to identify the genes up- and down-regulated in disease samples as compared to HC. The “common up” and “common down” gene sets at the disease level were identified by intersecting the upregulated or downregulated DEGs for COVID-19, sepsis, and SLE compared to HC. We performed Medical Subject Headings (MeSH) enrichment using the R package DOSE [82] for each cluster’s upregulated genes. We also calculated scores for the Molecular Signatures Database’s (MSigDB) signature gene set collection [83] using the R package GSVA (v.1.44.0) [84].

### 4.8. Calculation of Ligand-Receptor Interaction Scores between Platelets and Other Cell Types

Cell-cell ligand-receptor interactions were evaluated using the scoring system proposed by Kumar et al. [85]. The score uses ligand-receptor interactions from the iTALK ligand-receptor database [59]. The ligand-receptor interaction score between cell type A and cell type B is calculated as a product of the average receptors’ expression across all cells of cell type A and the average ligands’ expression across all cells of cell type B.
$$Interaction\;score(receptor,\;ligand,\;cell\;type\;A,\;cell\;type\;B)\;\;\;\;\;\;\;\;\;\;\;=\frac{1}{n_{cell\;type\;A}}\sum_{i\;\in \;cell\;type\;A}^{\;}\;e_{i,\;\;receptor}\times \frac{1}{n_{cell\;type\;B}}\sum_{j\;\in \;cell\;type\;B}^{\;}\;e_{j,\;\;ligand}$$

e_i,j_ = expression gene j in cell in_celltypeA_ = number of cells in the cell type A

### 4.9. Trajectory Inference of Transition in Platelets Subclusters

We presented the platelet trajectory analysis using monocle3 (v. 1.2.9) [86]. Trajectories were calculated, and the cells were displayed based on a monocle3 pseudotime approach rooted in the previously identified transitional platelets. Integrated gene expression matrices from each dataset were first exported from Seurat into monocle3 to construct a new_cell_data_set. The exported matrices were aligned using the single-cell batch correction method package batchelor (v. 1.12.0) [87]. Then, they were subjected to the standard PCA to preprocess the data, with the number of dimensions set at 100. The dimension reduction and clustering of cells were all set to the monocle3 default settings. Finally, we used reversed graph embedding to learn the principal graph from the reduced dimension space using the learn_graph function.

### 4.10. Data and Code Availability Statement

The published article includes all datasets generated or analyzed during this study. The Methods section provides details about the steps, functions, and parameters that were used for the analysis. Custom scripts for analyzing data are available at https://github.com/xqiu625/PlateletSubpop-ML-ScTranscriptomics (accessed on 28 May 2024). The sources of the data are provided in this paper.

## 5. Conclusions

In conclusion, this study identified distinct platelet subpopulations associated with disease severity and clinical outcomes, suggesting specific mechanisms of platelet disorders in COVID-19, sepsis, and SLE.

These subpopulations, such as the highly active coagulation cluster C4, the hypoxic cluster C9, and the quiescent cluster C11, are characterized by the presence of gene modules and pathways associated with survival or fatal outcomes, including those related to cytoplasmic translation, immune response, hypoxia, and coagulation. This list offers potential biomarkers and therapeutic targets for future investigation.

Our study also highlights common mechanisms of platelet dysfunction across infectious and autoimmune diseases, with shared alterations in pathways related to immune dysregulation, neurodegeneration, and cellular metabolism. These findings underscore the broader relevance of platelets as central players in the pathogenesis of inflammatory disorders and suggest potential avenues for transdiagnostic interventions.

Moreover, by analyzing platelet-immune cell interactions, we caught a glimpse into the complex interplay between platelets, monocytes, and lymphocytes in shaping the abnormal inflammatory state. Platelet-monocyte aggregation is seen to be linked with disease severity, and platelets are shown to play a role in lymphocyte activation and differentiation, highlighting the newly discovered immunomodulatory roles of platelets.

Adding machine learning, particularly XGBoost and deep neural networks, to the workflow of scRNAseq analysis demonstrates the power of these approaches to identify key features and biomarkers from high-dimensional single-cell data and their potential for clinical translation and personalized risk stratification.

In summary, our study represents a major step toward understanding the diverse roles of platelets in the abnormal inflammation characteristic of COVID-19, sepsis, and SLE. The integration of single-cell transcriptomics, machine learning, and cross-disease comparisons provides a powerful framework for uncovering novel biology and identifying promising therapeutic targets. 

## Figures and Tables

**Figure 1 ijms-25-05941-f001:**
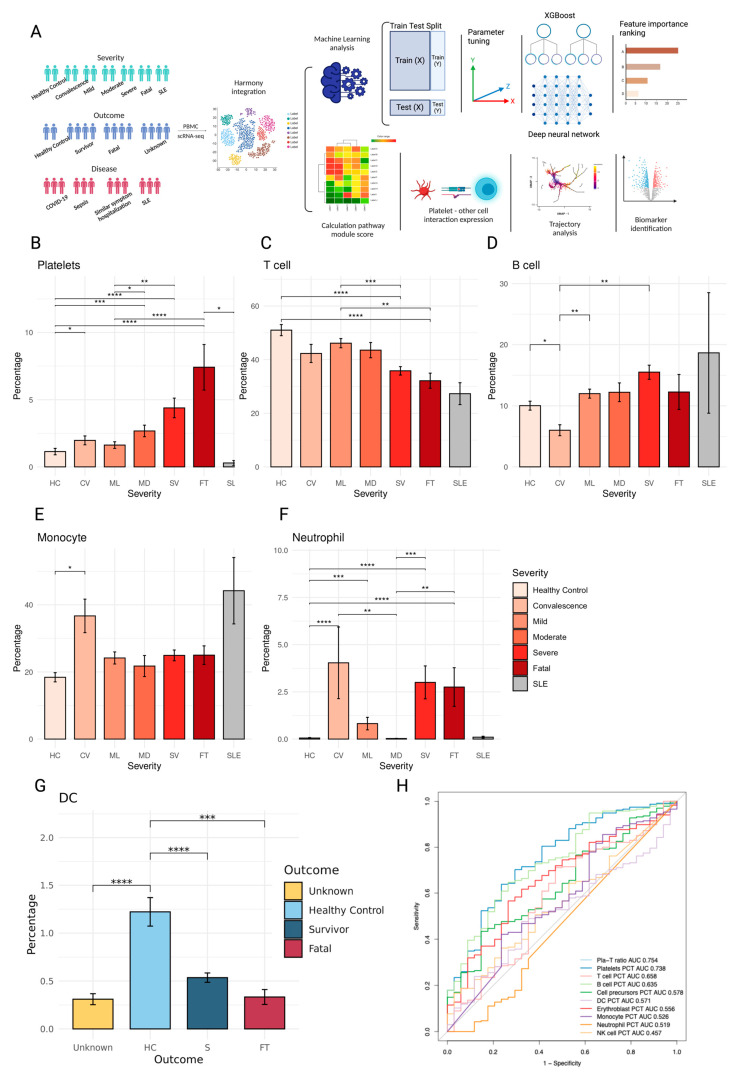
PBMC profiling from healthy controls, sepsis, similar symptom hospitalized, COVID-19, and SLE patients. (**A**) Schematic outline depicting the workflow for data collection from published literature and subsequent integrated analysis. Created with biorender.com. (**B**–**F**) Bar plots depicting the percentage of different cell types under different disease severities. (**B**) Platelets, (**C**) T cells, (**D**) B cells, (**E**) monocytes, and (**F**) neutrophils. (**G**) DC under different outcome situations. The differences in percentages associated with adjusted *p*-values below 0.05, 0.01, 0.001, and 0.0001 are indicated as *, **, ***, and ****, respectively, and not significant ones are not shown. The significance analysis was performed using Wilcoxon tests. Standard error bars were also added. (**H**) Receiver operating characteristic (ROC) curves for the platelet to T cell ratio and other cell type percentages were used to distinguish non-survivors from survivors.

**Figure 2 ijms-25-05941-f002:**
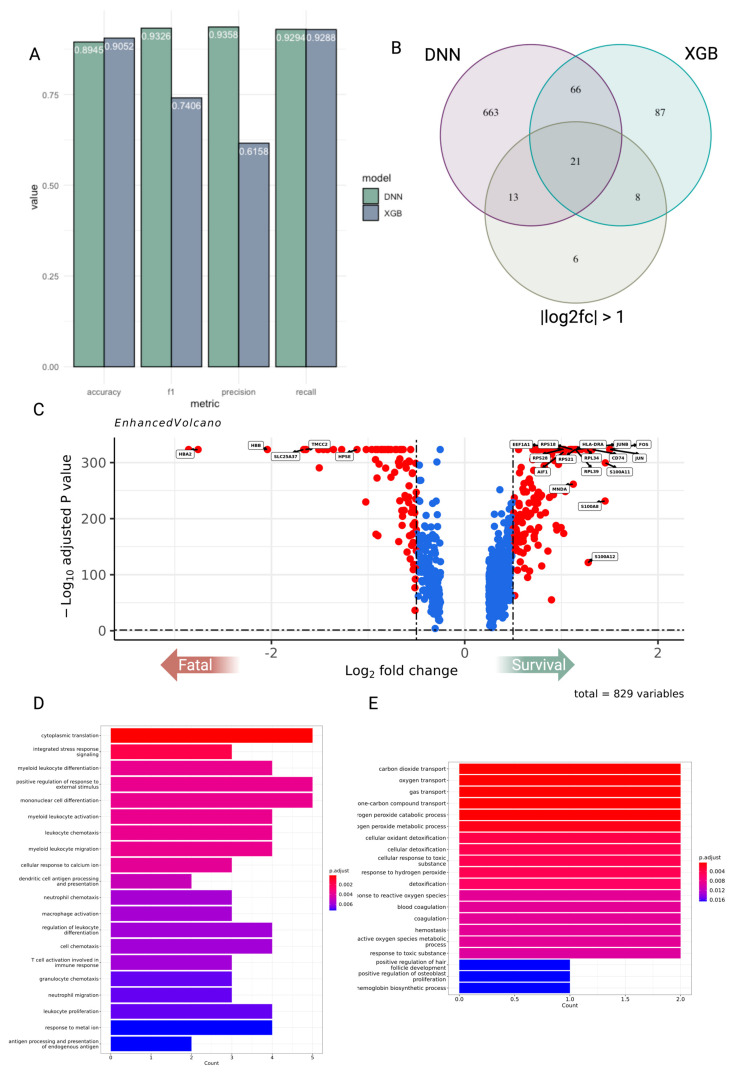
Deep neural networks and XGBoost modeling identify biomarkers of survival and fatal platelets. (**A**) Comparison of DNN and XGBoost Model Performance, DNN (represented in green) vs. XGB (represented in blue). (**B**) Venn diagram representing features from the Deep Neural Network (DNN) and XGBoost (XGB) models, specifically including only those features that rank in the highest 5% in terms of their importance or gain metrics within each model. Additionally, incorporate features that demonstrate a differential expression gene (DEG) profile with an absolute log2 fold change (log2fc) greater than 1. (**C**) Volcano plot depicting genes that are upregulated or downregulated when comparing platelets from survivors to those from fatal cases. The x-axis represents the log fold change. This is a measure of the change in expression levels of variables between two conditions. A zero value indicates no change, positive values indicate upregulation, and negative values indicate downregulation in the condition of interest relative to a reference condition. The y-axis represents the negative log10 adjusted *p*-value. This transformation is used to amplify differences in *p*-values, where small *p*-values (which indicate statistical significance) result in larger values on the plot. The horizontal dashed line typically represents a threshold of significance (e.g., adjusted *p*-value of 0.05), above which the findings are considered statistically significant. (**D**,**E**) Bar chart that illustrates the enrichment of certain biological pathways in a set of genes related to genes up-regulated in survival/fatal. The color of the bars represents the level of statistical significance after adjustment for multiple comparisons, with darker colors indicating more statistically significant enrichment.

**Figure 3 ijms-25-05941-f003:**
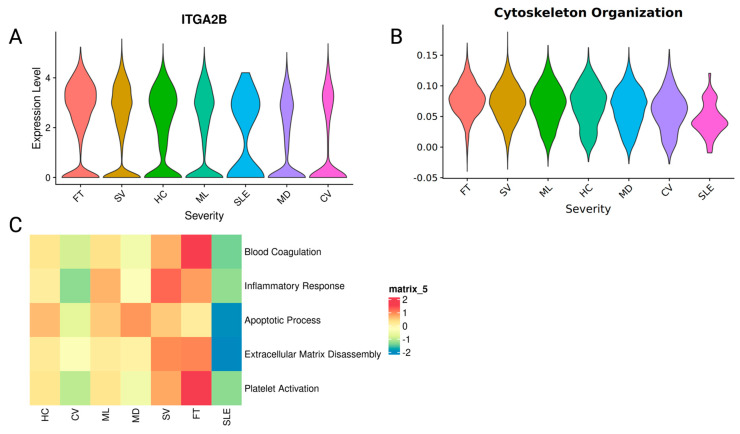
Differential expression of platelets affects endotheliopathy across disease severity states. (**A**) The expression of the ITGA2B gene in platelets across severity states. (**B**) The expression module GO: Cytoskeleton organization in platelets across severity states. Violin plots are ordered according to the decreasing average value of the expression. (**C**) The comparison of GO terms blood coagulation (GO:0007596), inflammatory response (GO:0006954), apoptotic process (GO:0006915), extracellular matrix disassembly (GO:0022617), and platelet activation (GO:0030168) expression. Heatmap coloring represents *z*-scored scores averaged across all cells in a given sample.

**Figure 4 ijms-25-05941-f004:**
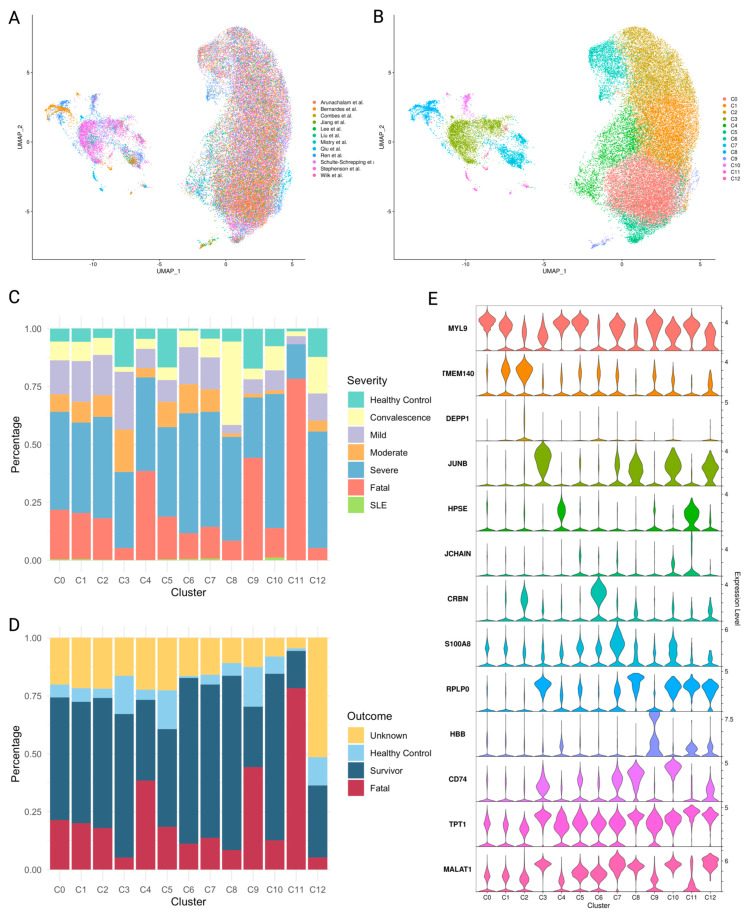
Clustered integrative analysis of platelets’ single-cell transcriptional landscape. (**A**,**B**) Cell cluster UMAP representation of all merged platelets (**A**) colored by data source; (**B**) colored by clusters. (**C**,**D**) stacked bar plots display platelet cluster proportion under (**C**) different disease severity, and (**D**) different outcome situations. (**E**) Violin plots showing expression levels of marker genes for each cluster in platelets. In (**A**), we gathered single-cell RNA-seq datasets of peripheral blood mononuclear cells (PBMCs) from COVID-19 [30,31,32,33,34,35,36,37,38], sepsis [12,39], and systemic lupus erythematosus (SLE) [40] patients.

**Figure 5 ijms-25-05941-f005:**
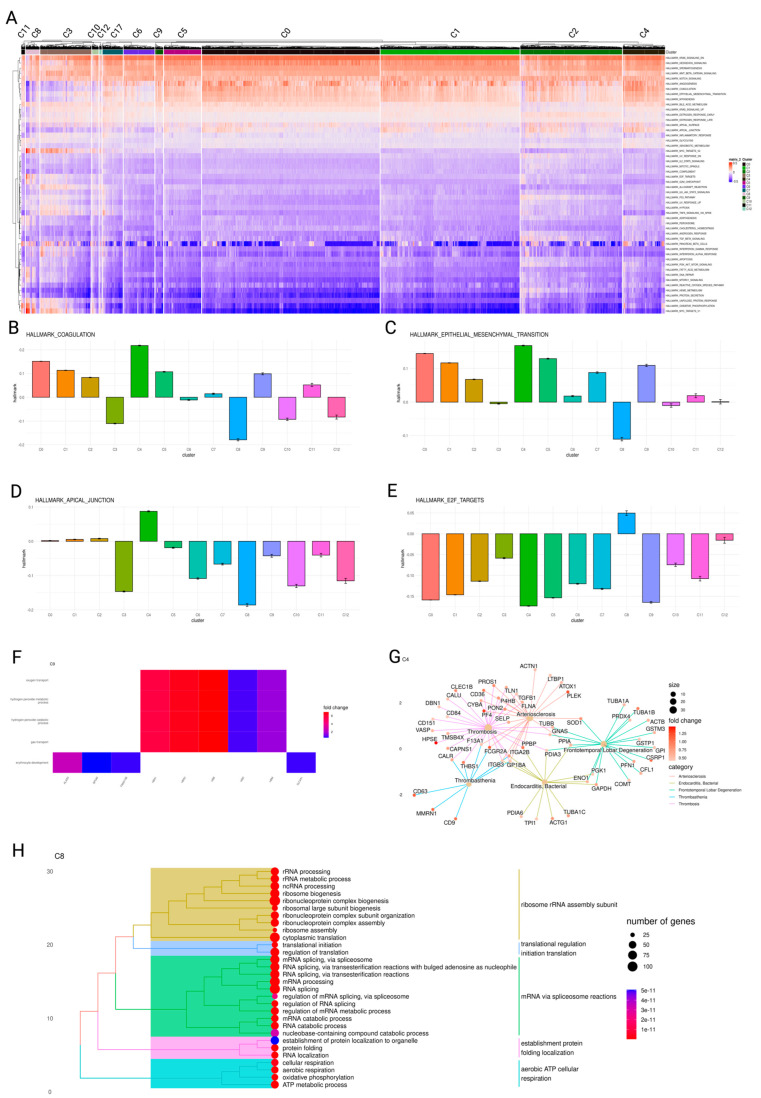
Clustered platelets and their unique pathway expression changes. (**A**) Enrichment analysis for human hallmark gene sets for each platelet cluster. Expression level is the normalized enrichment score in the GSEA algorithm. (**B**–**E**) Bar plot of hallmark gene sets expression among clusters. (**B**) Coagulation (**C**) Apical Junction (**D**) Epithelial Mesenchymal Transition (**E**) E2F Targets. (**F**) Heatmap display fatal cluster C9 up-regulated genes and enriched pathways in gene ontology (GO). (**G**) Gene-Concept network display fatal cluster C4 up-regulated genes and enriched pathways from Gene, Disease Features Ontology-based Overview System (gendoo) [41] and diseases category. (**H**) Tree plot display convalescence cluster C8 enriched pathways in GO.

**Figure 6 ijms-25-05941-f006:**
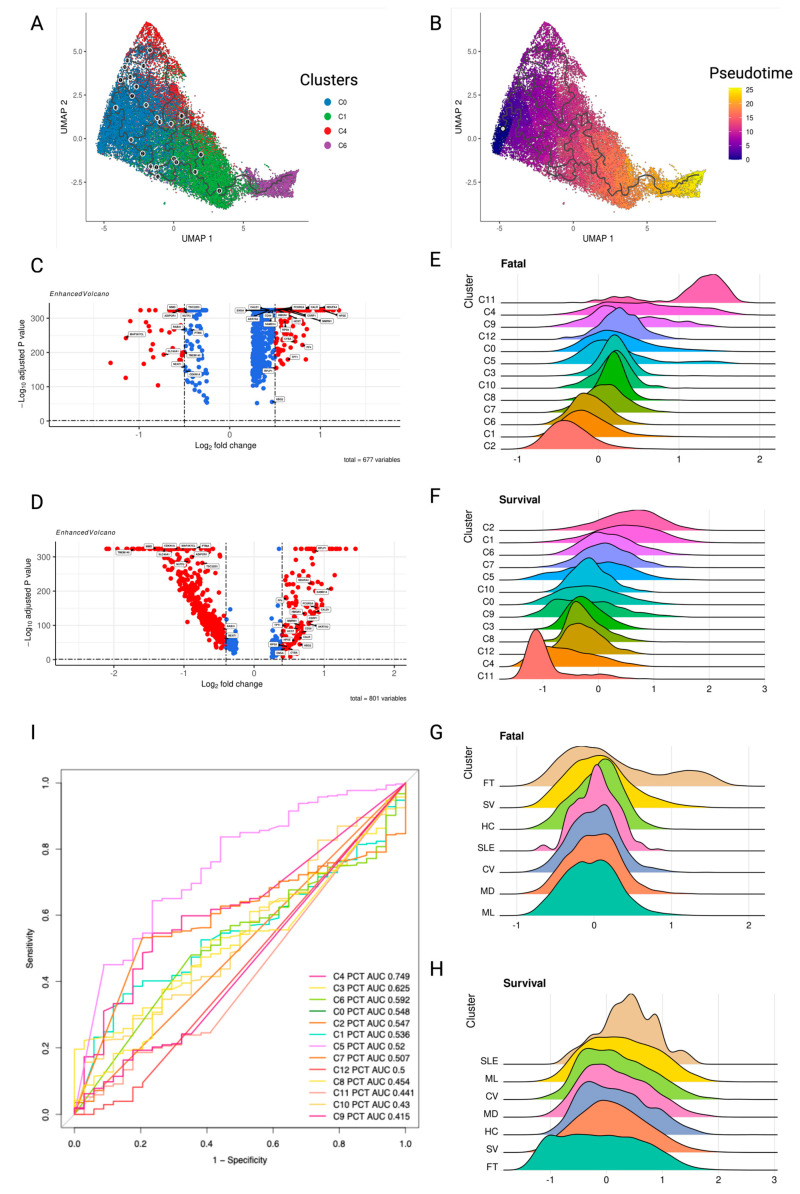
Platelet module signatures in patients related to survival and the fatal dynamic trend (**A**,**B**) Pseudo-time plot of platelets from clusters C0, C1, C4, and C6 exhibiting trajectory fates. (**C**,**D**) Differential expression genes in (**C**) C4 vs. C0; (**D**) C0 vs. C1. The volcano plot displays the genes constantly up-or down-regulated in the direction of disease severity. Volcano plots were prepared with the R package EnhancedVolcano (v.1.13.2). In the volcano plot, genes with an absolute log2 fold change greater than 0.5 are represented by red dots, while those with a lower absolute log2 fold change are represented by blue dots. (**E**–**H**) Ridge plots showing the density of expression level of (**E**) Fatal module expression under platelet clusters, (**F**) Fatal module expression under disease severities, (**G**) Survival module expression under platelet clusters, and (**H**) Survival module expression under disease severities. Ridge plots are ordered in descending order. (**I**) Receiver operating characteristic (ROC) curves for each platelet cluster percentage from PBMC were used to distinguish non-survivors from survivors.

**Figure 7 ijms-25-05941-f007:**
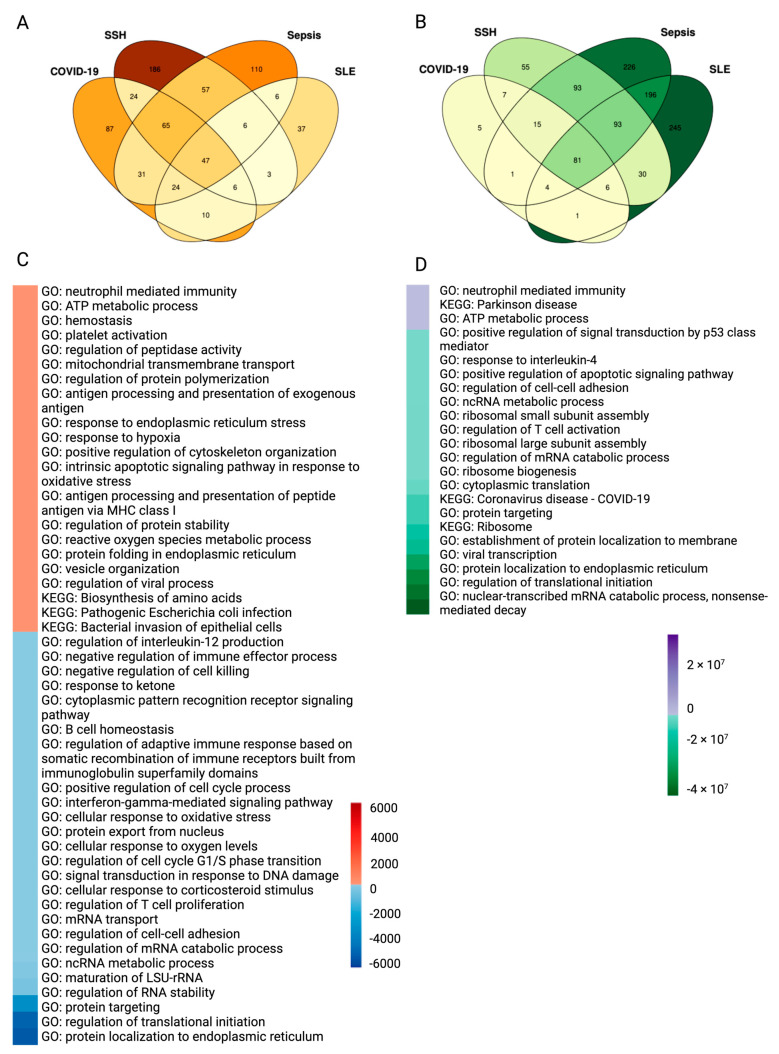
Platelets pathway expression among healthy controls, sepsis, similar symptom hospitalized, COVID-19, and SLE patients. (**A**,**B**) Venn diagrams describing changes in Gene Ontology (GO) and Kyoto Encyclopedia of Genes and Genomes (KEGG) pathways from COVID-19, similar symptoms hospitalized (SSH), sepsis, and systemic sclerosis lupus (SLE) vs. healthy controls (HC), (**A**) Up-regulated pathways. (**B**) Down-regulated pathways. The pathways were filtered for those with an adjusted *p*-value under 0.05. (**C**) Heatmap illustration of non-survivor vs. survivor up- or down-regulated pathways. Colors are decided by the product of the COVID-19 and sepsis up/down-regulated enriched pathway log10 (adjusted *p* value). The GO terms were reduced to representative ones using R package rrvgo (v.1.8.0) [44] (the cutoffs were similarity > 0.4) and then overlapped. (**D**) Heatmap illustration of diseases vs. health controls up- or down-regulated pathways. Color is decided by the product of the COVID-19, SSH, SLE, and sepsis up- or down-regulated enriched pathway log10 (adjusted *p* value). The GO terms were reduced to representative ones using R package rrvgo (1.8.0) (the cutoffs were similarity > 0.1) and then overlapped.

**Figure 8 ijms-25-05941-f008:**
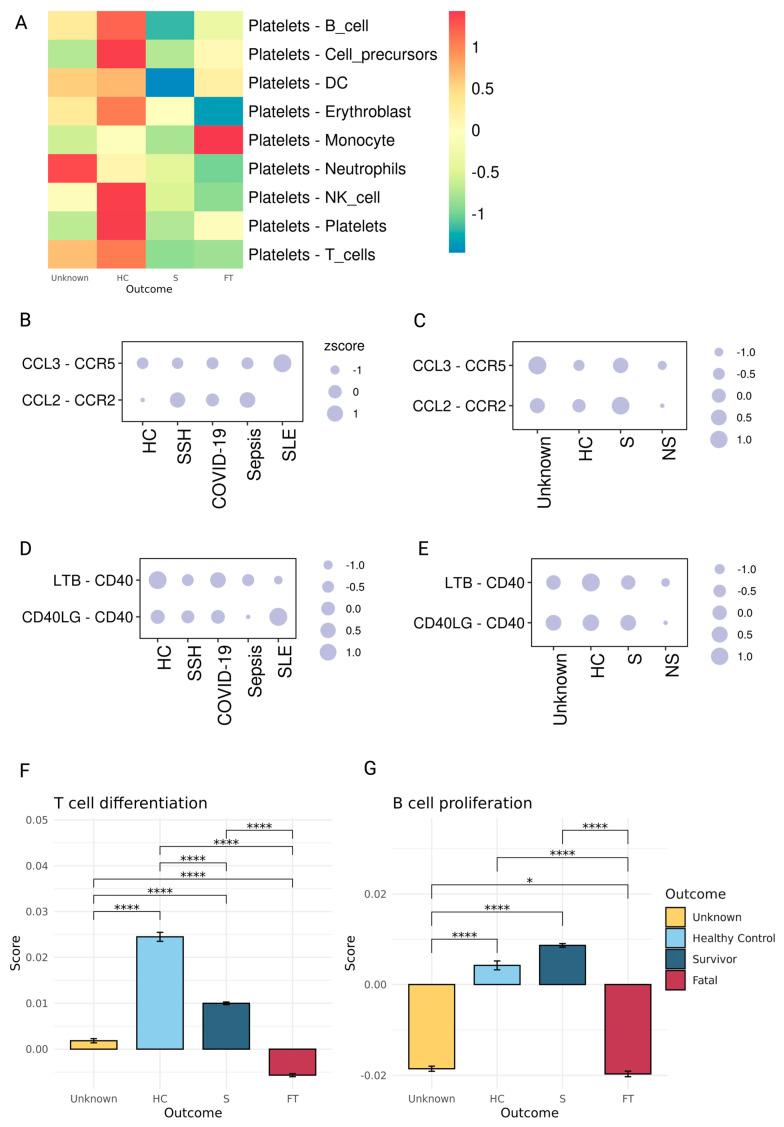
Alterations in platelet and other cell type interactions among healthy controls, sepsis, similar symptom hospitalized, COVID-19, and SLE patients. (**A**) Comparison of ligand-receptor interaction scores between platelets and other cell types. Heatmap coloration corresponds to z-scored, log-normalized mean interaction scores averaged across all cells from a specific sample. (**B**–**E**) Ligand and receptor interaction scores between platelets and (**B**) T cells across various disease severity levels, (**C**) T cells across different outcomes, (**D**) B cells across various disease severity levels, and (**E**) B cells across different outcomes. The circle size represents the *z*-scored interaction scores. (**F**,**G**) Pathway module scores across different outcome situations in platelets, including (**F**) T cell differentiation and (**G**) B cell proliferation. The differences in scores associated with adjusted P-values below 0.05, 0.01, 0.001, and 0.0001 are indicated as * and ****, respectively.

**Table 1 ijms-25-05941-t001:** Platelet transcriptional alterations across varying disease severity levels. The scores in each cell were calculated as the mean expression level of all genes within a particular module; each gene in the module was quantified, and the expression level was averaged to get a single score that represents the overall activity of that pathway in the severity condition.

Pathway Module	Healthy Control	Convalescence	Mild	Moderate	Severe	Fatal	SLE
Glycolysis	0.0221	0.0585	0.0151	0.0065	0.0131	0.0736	−0.0675
MHC ClassII	−0.0319	0.0672	−0.0318	−0.0438	−0.0575	−0.1183	−0.0519
OXPHOS	0.0323	0.0419	0.0191	0.047	0.0474	0.0801	0.0469
Coagulation	0.3007	0.2432	0.3034	0.2642	0.3243	0.3643	0.2209
Response to Interferon Beta	−0.1375	−0.0454	−0.0335	0.0277	−0.0373	−0.0263	−0.0674
Response to Interferon Gamma	0.094	0.1052	0.1109	0.1293	0.1024	0.1091	0.0603
Response to Type I Interferon	−0.0947	−0.0314	−0.0091	0.0364	−0.0245	−0.0201	−0.0045
Translational Initiation	0.0877	0.0778	0.0653	0.0841	0.0549	0.0535	0.058

**Table 2 ijms-25-05941-t002:** Platelet numbers analyzed from single-cell RNA-seq datasets of peripheral blood mononuclear cells (PBMCs) of COVID-19, sepsis, and systemic lupus erythematosus (SLE) patients categorized according to disease severity, outcomes, and diseases.

Group	Platelets
Healthy Controls (HC)	3205
Convalescence (CV)	3695
Mild (ML)	7359
Moderate (MD)	4330
Severe (SV)	19,805
Fatal (FT)	9414
Systemic Lupus Erythematosus (SLE)	169
Outcomes	
Fatal (FT)	9414
Survivors (S)	25,750
Unknown	9608
Diseases	
COVID-19	38,673
Hospitalized Patients with Similar Symptoms (SSH)	2508
Sepsis	3422
Systemic Lupus Erythematosus (SLE)	169

## Data Availability

The published article includes all datasets generated or analyzed during this study. The Methods section provides details about the steps, functions, and parameters that were used for the analysis.

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
