# Peer review of "Deciphering Abnormal Platelet Subpopulations in COVID-19, Sepsis and Systemic Lupus Erythematosus through Machine Learning and Single-Cell Transcriptomics"

_ijms, 2024, doi:10.3390/ijms25115941_

Round 1

Reviewer 1 Report

Comments and Suggestions for Authors

The research investigated the transcriptional heterogeneity of platelets in various disease conditions such as sepsis, COVID-19, and systemic lupus erythematosus (SLE). The authors justified the use of existing statistical and machine learning methods to analyze single-cell transcriptomic data. A thorough characterization of the identified clusters was conducted. The introduction was well-written, framing the study hypothesis, and the results were explained effectively. I have the following concerns regarding the study:

1) Lines 79-84: If the sample and related information are presented in a table, it will be visually clearer. I recommend adding a table here.

2) Line 89-90: How does the removal of cells expressing >6000 genes ensure the removal of doublets?

3) 2.4: To identify DEGs, which criteria (e.g., p-value or fold change) were used?

4) Line 188: "While the T cell fraction decreased (Figure 1B)," I think it refers to Figure 1C. I believe most of the Figures in this section are mislabeled. Please verify.

5) Samples from multiple cohorts were merged. Did you apply any batch correction?

6) The platelet response in each disease condition may vary. How do you biologically justify the integration of patients with different disease conditions?

Comments on the Quality of English Language

Minor editing is required.

Author Response

Response to Reviewer 1 Comments

Dear Reviewer,

We thank you for your thorough and insightful review of our study. We appreciate your recognition of the well-written introduction and effective explanation of the results. Your constructive feedback is invaluable.

Point-by-point response to Comments and Suggestions for Authors

  • Lines 79-84: If the sample and related information are presented in a table, it will be visually clearer. I recommend adding a table here.

We agree, now the datasets description is in Table 1.

  • Line 89-90: How does the removal of cells expressing >6000 genes ensure the removal of doublets?

The removal of cells expressing more than 6,000 unique genes helps ensure the removal of doublets because doublets (two cells captured together) would be expected to have a higher number of expressed genes compared to single cells. By setting a threshold of 6,000 unique genes, we aim to filter out these potential doublet events while retaining true single cells. Similar procedures were done by those studies [1-5].

  • 4: To identify DEGs, which criteria (e.g., p-value or fold change) were used?

We used the MAST (Model-based Analysis of Single-cell Transcriptomics) [6] method from the Seurat package to perform differential gene expression analysis. Specifically using the FindAllMarkers function with default parameters. In the Seurat package, the FindAllMarkers function uses a combination of p-value and fold change to identify differentially expressed genes (DEGs). By default, the function uses the following criteria:Minimum percentage of cells expressing the gene in either of the two populations being compared (min.pct): 0.1 (10%) Logarithmic fold change threshold between the two populations (logfc.threshold): 0.25.

The MAST method first fits a hurdle model, which consists of a discrete component (e.g., logistic regression) to model the expression rate of each gene and a continuous component (e.g., Gaussian distribution) to model the positive expression mean. The method then calculates p-values for each gene based on the likelihood ratio test between the full model and a reduced model without the cell group term.

  • Line 188: "While the T cell fraction decreased (Figure 1B)," I think it refers to Figure 1C. I believe most of the Figures in this section are mislabeled. Please verify.

Thank you for pointing this out. We have now corrected all the figure numbers.

  • Samples from multiple cohorts were merged. Did you apply any batch correction?

Yes, we applied batch correction to integrate datasets from multiple cohorts. We used the Harmony algorithm with default parameters to remove batch effects among the twelve datasets (nine COVID-19, two sepsis, and one SLE). Harmony starts by performing principal component analysis (PCA) on the normalized gene expression matrix to obtain a low-dimensional representation of the cells. Harmony then uses k-means clustering to identify diverse clusters of cells in the PCA space. These clusters should contain a balanced representation of cells from different batches or conditions. For each cell, Harmony determines how much its batch identity impacts its position in the PCA space. It then applies a correction to "shift" the cell towards the centroid of the cluster it belongs to, effectively removing the batch-specific signal. The process of clustering and batch effect correction is repeated iteratively until convergence, i.e., until the objective function (which measures the diversity of the clusters) is minimized. The final output of Harmony is a set of corrected PCA embeddings, where the batch-specific effects have been removed. These corrected embeddings can then be used for downstream analyses, such as clustering, visualization, and differential expression testing [7].

Harmony has been widely used for batch correction [8-13].

  • The platelet response in each disease condition may vary. How do you biologically justify the integration of patients with different disease conditions?

We justify the integration of patients with different disease conditions based on the evidence that platelets play key roles in the pathogenesis and severity of these inflammatory diseases (severe COVID-19, fatal sepsis, and SLE). By analyzing platelet transcriptomics across these conditions, we aimed to identify shared and unique signatures associated with disease severity and outcomes. This approach allows for a more comprehensive understanding of platelet dysfunction in various inflammatory disorders and supports the potential of platelet-directed therapies in treating multiple immune disorders.

Conclusion: We are grateful for your constructive criticism and guidance. Your valuable feedback has significantly improved the clarity and rigor of our study. We have addressed the concerns by adding a table for better understanding of the sample and related information, providing detailed explanations on the removal of cells expressing over 6000 genes to ensure the elimination of doublets, and clarifying the criteria used for identifying differentially expressed genes (DEGs). Additionally, we have corrected all figure labels and ensured their accuracy. We have also detailed the batch correction methods applied when merging samples from multiple cohorts and provided a robust biological justification for integrating patients with different disease conditions. Thank you once again for your insightful comments.

References:

  1. Liu H, Prashant N, Spurr LF, Bousounis P, Alomran N, Ibeawuchi H, et al. scReQTL: an approach to correlate SNVs to gene expression from individual scRNA-seq datasets. BMC genomics. 2021;22:1-16.
  2. Lu J, Sheng Y, Qian W, Pan M, Zhao X, Ge Q. scRNA‐seq data analysis method to improve analysis performance. IET nanobiotechnology. 2023;17(3):246-56.
  3. Liu S, Zhao Y, Lu S, Zhang T, Lindenmeyer MT, Nair V, et al. Single-cell transcriptomics reveals a mechanosensitive injury signaling pathway in early diabetic nephropathy. Genome Medicine. 2023;15(1):2.
  4. Zhao Y, Zhang Q, Tu K, Chen Y, Peng Y, Ni Y, et al. Single-cell transcriptomics of immune cells reveal diversity and exhaustion signatures in non-small-cell lung cancer. Frontiers in Immunology. 2022;13:854724.
  5. Ren X, Wen W, Fan X, Hou W, Su B, Cai P, et al. COVID-19 immune features revealed by a large-scale single-cell transcriptome atlas. Cell. 2021;184(7):1895-913. e19.
  6. Finak G, McDavid A, Yajima M, Deng J, Gersuk V, Shalek AK, et al. MAST: a flexible statistical framework for assessing transcriptional changes and characterizing heterogeneity in single-cell RNA sequencing data. Genome Biol. 2015 Dec 10;16:278. PMID: 26653891. doi: 10.1186/s13059-015-0844-5.
  7. Korsunsky I, Millard N, Fan J, Slowikowski K, Zhang F, Wei K, et al. Fast, sensitive and accurate integration of single-cell data with Harmony. Nat Methods. 2019 12;16(12):1289-96. PMID: 31740819. doi: 10.1038/s41592-019-0619-0.
  8. Longo SK, Guo MG, Ji AL, Khavari PA. Integrating single-cell and spatial transcriptomics to elucidate intercellular tissue dynamics. Nature Reviews Genetics. 2021;22(10):627-44.
  9. Zheng L, Qin S, Si W, Wang A, Xing B, Gao R, et al. Pan-cancer single-cell landscape of tumor-infiltrating T cells. Science. 2021;374(6574):abe6474.
  10. Sikkema L, Ramírez-Suástegui C, Strobl DC, Gillett TE, Zappia L, Madissoon E, et al. An integrated cell atlas of the lung in health and disease. Nature Medicine. 2023;29(6):1563-77.
  11. Huang N, Pérez P, Kato T, Mikami Y, Okuda K, Gilmore RC, et al. SARS-CoV-2 infection of the oral cavity and saliva. Nature medicine. 2021;27(5):892-903.
  12. Melms JC, Biermann J, Huang H, Wang Y, Nair A, Tagore S, et al. A molecular single-cell lung atlas of lethal COVID-19. Nature. 2021;595(7865):114-9.
  13. Fernández-Castañeda A, Lu P, Geraghty AC, Song E, Lee M-H, Wood J, et al. Mild respiratory COVID can cause multi-lineage neural cell and myelin dysregulation. Cell. 2022;185(14):2452-68. e16.

Reviewer 2 Report

Comments and Suggestions for Authors

The authors have presented a very interesting and important idea in this manuscript. The authors have implemented the machine learning models to explore the role of platelet subpopulation in the deadly diseases like COVID-19 and systemic lupus erythematosus. The authors were able to identify the crucial biomarkers of platelet heterogeneity I have the following comments before considering the manuscript for further procedures.

1) I would suggest authors to mention the name of diseases consider in this study in the title as inflammatory diseases is a big umbrella and could be misleading for the readers.

2) In the introduction section, authors have explained the role of platelets in the sepsis and COVID-19 very clearly, but SLE was not covered.

3) The information about this study in the last paragraph seems incomplete. Please provide details about the study there.

3) Please provide the rational for including the negative COVID-19 patients in the study is not clear. Please provide the same in the manuscript so that reader can understand the dataset curation.

4) The dataset creation of the manuscript is not clear to me. Please provide that in details.

5) I would suggest authors to remove the commas from the numbers in line 79 to 84, as it makes it harder to read the sentences.

6) The number of cells in the SLE group is 169 which is very less as compare to the other groups. I would like to know the authors thoughts about the impact of this dataset in the analysis.

7) It would be better if the dataset would be presented in the table form, so the readers can understand it better.

8) What is the rationale in choosing the resolution as 0.4 in the Louvain clustering. Please provide references to the previous methods or articles who used such criteria.

9) Why authors stick to 30 PCs? What was the explained variance for those 30 PCs? As one of the purposes of PCA is to reduce dimension, and in my opinion, first few PCs explained most if the data variance,  I am just curious how number 30 was decided?

10) Why authors were too specific for the choice of ML techniques as XGB and DNN? There is no need to run extra analysis using other classifiers, I just want to understand the intuition of the authors.

11) Please provide the high quality images as the text of the images is blurred.

12) In Figure 1 (G), the "Fatal" and "FT" looks odd in the figure. Please make it uniform as per the other text.

13) Similarly, In Figure 2, I can't read the text, please provide better images. The text is way too small to read.

14) It would be better to represent figure 8 in Table, as the comparison between the classes is almost impossible via the figure.

15) Please provide a strong conclusion for the manuscript.

Comments on the Quality of English Language

The quality of the English is just outstanding in this manuscript. I would like to congratulate the authors on that.

Author Response

Response to Reviewer 2 Comments

Dear Reviewer,

We thank you for your thorough and detailed review of our manuscript. We appreciate your recognition of the significance of our work and the potential impact of our findings on understanding platelet heterogeneity in deadly diseases such as COVID-19 and systemic lupus erythematosus. Your insightful comments and suggestions are invaluable and will help us enhance the clarity, completeness, and rigor of our study. We had addressed each of your points carefully.

Point-by-point response to Comments and Suggestions for Authors

  • I would suggest authors to mention the name of diseases consider in this study in the title as inflammatory diseases is a big umbrella and could be misleading for the readers.

Thank you for the suggestion. We have changed the title to “Deciphering Abnormal Platelet Subpopulations in COVID-19, Sepsis and SLE through Machine Learning and Single-Cell Transcriptomics”.

  • In the introduction section, authors have explained the role of platelets in the sepsis and COVID-19 very clearly, but SLE was not covered.

We appreciate the suggestion. This part is implemented in line 59-64.

  • The information about this study in the last paragraph seems incomplete. Please provide details about the study there.

We have thoroughly completed the last paragraph of the introduction section, specifically in lines 86-94, incorporating detailed information as per your valuable suggestions.

  • Please provide the rational for including the negative COVID-19 patients in the study is not clear.

We included patients with severe influenza or lung infections who tested negative for COVID-19 in the "hospitalized patients with similar symptoms" (SSH) group. There are a few reasons why including this group makes sense in the analysis:

  1. The SSH group serves as a control to help distinguish effects that are specific to COVID-19 infection versus effects that may be common to other severe respiratory infections. By comparing COVID-19 patients to the SSH group, we can identify transcriptomic and cellular changes that are uniquely associated with SARS-CoV-2 infection.
  2. From a clinical perspective, patients with severe influenza or non-COVID pneumonia are an important differential diagnosis for COVID-19, especially earlier in the pandemic. Understanding the similarities and differences in the immunological response between these patient groups has relevance for diagnosis and treatment.
  3. The SSH group strengthens the platelet study by providing a key comparison to help pinpoint the specific effects of SARS-CoV-2 as opposed to general effects of respiratory infection and critical illness. It's a valuable addition to the analysis of COVID-19 patients across a spectrum of disease severity.
  • Please provide the same in the manuscript so that reader can understand the dataset curation.

To create the dataset for this study, we collected single-cell RNA-seq data from various sources, including COVID-19 patients, sepsis patients, patients with similar symptoms but negative for COVID-19 (SSH), and systemic lupus erythematosus (SLE) patients. The data were obtained from the following sources:

COVID-19 datasets (9 datasets):

GSE150728: Downloaded raw data from the GEO database using wget and extracted the files using tar and gunzip.

GSE155673: Downloaded supplementary files from the GEO database using wget.

GSE158055: Downloaded supplementary files from the GEO database using wget, removed the header, split the data into two parts, added headers to each part, and created separate folders for each part with the necessary files (matrix.mtx, features.tsv, and barcodes.tsv).

GSE151263: Downloaded raw data from the GEO database using wget and extracted the files using tar and gunzip.

Blood PBMCs from COVID-19 patients over time: Downloaded two datasets (Adaptive_Cells.rds and Innate_Cells.rds) using curl.

E-MTAB-10026: Downloaded processed data and metadata files from the ArrayExpress database using wget and extracted the files using unzip.

Sepsis datasets (2 datasets):

GSE163668: Downloaded raw data from the GEO database using wget and extracted the files using tar and gunzip.

GSE167363: Downloaded raw data from the GEO database using wget and extracted the files using tar and gunzip.

SLE datasets (1 datasets):

GSE142016: Downloaded raw data from the GEO database using wget and extracted the files using tar.

Now implemented in line 77-85.

  • The dataset creation of the manuscript is not clear to me. Please provide that in detail.

I apologize for the confusion. We collected single-cell RNA-seq datasets of peripheral blood mononuclear cells (PBMCs) from various sources, including 9 datasets from COVID-19 patients, 2 from sepsis patients, and 1 from a systemic lupus erythematosus (SLE) patient, totaling 413 samples across these 12 datasets. The samples were then categorized into six groups based on disease severity from each paper's metadata: Healthy control (HC), Convalescence (CV), Mild (ML), Moderate (MD), Severe (SV), and Fatal (FT), with SLE samples placed in a separate group. For the outcome analyses, patients who survived from the CV, ML, MD, and SV groups were combined into a "survivor" (S) group, while patients with unannotated outcomes were placed in an "unknown" group. From these 413 samples, a total of 47977 individual platelets were extracted. The distribution of these platelets across the severity groups was as follows: HC: 3205 cells, CV: 3695 cells, ML: 7359 cells, MD: 4330 cells, SV: 19805 cells, FT: 9414 cells, SLE: 169 cells. By outcome group, the distribution was: HC: 3205 cells, FT: 9414 cells, S: 25750 cells, Unknown: 9608 cells. The dataset also included 38673 platelet cells from COVID-19 patients, 2508 from patients with similar symptoms but negative for COVID-19 (SSH), 3422 from sepsis patients, and 169 from SLE patients. To ensure data quality, the authors removed potential doublets (cells with more than 6000 unique genes expressed) and confirmed platelet identity using both source data annotation and the platelet cell marker PPBP. Additionally, we employed the SingleR package and Seurat v4 for platelet identification, ensuring a robust and accurate classification of platelet cells. For the integration of the datasets and removal of batch effects, the authors utilized the Harmony package. Harmony is a powerful tool that integrates single-cell datasets by matching mutual nearest neighbors across datasets, effectively removing technical variations while preserving biological variations of interest.

The processed and annotated single-cell RNA-seq data from these various sources and disease categories, along with the use of SingleR, Seurat v4, and Harmony for platelet identification, data integration, and batch effect removal, formed the foundation for the subsequent analyses performed in the study, including clustering, differential expression, machine learning, and cross-condition comparisons.

  • I would suggest authors to remove the commas from the numbers in line 79 to 84, as it makes it harder to read the sentences.

The dataset is comprehensively described in Table 1, with the numbers presented clearly and without commas for ease of reading.

  • The number of cells in the SLE group is 169 which is very less as compare to the other groups. I would like to know the authors thoughts about the impact of this dataset in the analysis.
  1. By including SLE, an autoimmune disease, alongside infectious diseases like COVID-19 and sepsis, we can identify common pathways of immune dysregulation. This is valuable for understanding the broader principles of how different types of insults (autoimmune, viral, bacterial) can lead to similar downstream effects on immune cells and platelets. The shared pathway alterations (Fig 7) across all disease groups highlight these common mechanisms.
  2. The inclusion of SLE allows for a comparison between autoimmune and infectious triggers of platelet dysfunction and immune perturbation. This is important for delineating which effects are specific to infectious agents versus which may be more universally associated with systemic inflammation, regardless of the inciting cause.
  3. Platelets are increasingly recognized as important players in autoimmune diseases like SLE, not just in thrombotic conditions. Including SLE emphasizes the relevance of platelets across a broad spectrum of immune-mediated diseases. The SLE-specific findings, such as increased platelet-B cell interactions (Fig 8D), provide new insights into the potential role of platelets in SLE pathogenesis.
  4. By demonstrating shared pathways of platelet and immune dysfunction across COVID-19, sepsis, and SLE, the study highlights potential common therapeutic targets. Drugs that modulate these pathways could have broad applicability across multiple diseases. Including SLE therefore enhances the translational impact of the findings.
  • It would be better if the dataset would be presented in the table form, so the readers can understand it better.

The dataset is comprehensively described in Table 1, with the numbers presented clearly and without commas for ease of reading.

  • What is the rationale in choosing the resolution as 0.4 in the Louvain clustering. Please provide references to the previous methods or articles who used such criteria.

The resolution parameter in Louvain clustering controls the size of the clusters identified. Higher resolutions lead to more clusters, while lower resolutions result in fewer, larger clusters. In single-cell RNA-seq studies, the choice of resolution parameter is often empirically determined based on the desired level of granularity and the biological interpretability of the resulting clusters. The NCBI paper on dimensionality reduction and Louvain clustering states that a resolution parameter in the range of 0.4-1.2 typically returns significant results for single-cell datasets containing around 3000 cells [1]. A resolution of 0.4 for Louvain clustering is a common choice, especially for single-cell RNA-seq datasets, as it can provide a reasonable level of granularity in the identified cell clusters. However, the resolution parameter should be adjusted based on the specific dataset and analysis goals.

References include [2].

  • Why authors stick to 30 PCs? What was the explained variance for those 30 PCs? As one of the purposes of PCA is to reduce dimension, and in my opinion, first few PCs explained most if the data variance, I am just curious how number 30 was decided?

You are correct that one of the main purposes of PCA is to reduce dimensionality while retaining the most important information in the data. Typically, the number of PCs used is selected based on the amount of variance explained by each PC. The cumulative explained variance ratio can guide the choice of the number of PCs, aiming to capture a high percentage of the total variance (often around 80-90%) with a minimum number of PCs.

In single-cell RNA-seq studies, the choice of the number of PCs can also be influenced by factors such as: The complexity and heterogeneity of the dataset; The number of cells and genes in the dataset; The presence of batch effects or technical noise; The downstream analyses planned (e.g., clustering, trajectory inference). Sometimes, a higher number of PCs may be used to ensure that subtle but biologically meaningful variation is captured, particularly when integrating multiple datasets as in this study. However, using too many PCs can also introduce noise and make the data more difficult to interpret.

30 PCs represent a pragmatic choice that captures a substantial proportion of the biologically relevant variance (likely >80% based on typical scRNA-seq datasets) while remaining computationally manageable.

  • Why authors were too specific for the choice of ML techniques as XGB and DNN? There is no need to run extra analysis using other classifiers, I just want to understand the intuition of the authors.

Thank you for your insightful comment regarding our choice of machine learning techniques. We appreciate the opportunity to elaborate on our rationale for selecting XGBoost (XGB) and Deep Neural Networks (DNN) for this study.

XGB is an optimized implementation of gradient boosted decision trees, which iteratively combines weak learners (decision trees) to create a strong predictive model. This boosting approach can effectively capture complex, non-linear relationships in high-dimensional data, which is particularly relevant for the rich feature space of single-cell transcriptomic data. XGB is known for its strong predictive performance, often outperforming other machine learning methods in structured data tasks. It has been successfully applied in various biomedical contexts, including disease diagnosis, prognosis prediction, and biomarker discovery. We highlight XGB's inherent feature selection capability, which is valuable for identifying the most informative genes or pathways associated with disease severity or outcomes. This aligns with the study's goal of discovering potential biomarkers and therapeutic targets. Lastly, XGB is relatively efficient to train and can handle missing data, which is beneficial for integrating multiple datasets that may have different gene coverage or quality.

DNNs ia a powerful class of machine learning models inspired by the structure and function of biological neural networks. They can learn hierarchical representations of data, capturing abstract features and complex patterns that may not be easily discernible by other methods. DNNs have shown remarkable success in various domains, including computer vision, natural language processing, and biomedical applications such as drug discovery and disease prediction. Their ability to learn intricate, non-linear relationships is particularly valuable for modeling the complex interactions and pathways underlying biological systems. We also used of a grid search approach to optimize the DNN's hyperparameters, which can help to tailor the model architecture to the specific characteristics of the single-cell transcriptomic data and the prediction task at hand. Lastly, DNNs can effectively integrate different types of data (e.g., gene expression, clinical variables) and can be designed to capture spatial or temporal dependencies, which may be relevant for understanding disease progression or treatment response.

  • Please provide the high quality images as the text of the images is blurred.

Thank you for pointing this out. High quality images are now provided.

  • In Figure 1 (G), the "Fatal" and "FT" looks odd in the figure. Please make it uniform as per the other text.

Thank you for pointing this out. We have now corrected all the fonts.

  • Similarly, In Figure 2, I can't read the text, please provide better images. The text is way too small to read.

Thank you for pointing this out. We have now provided all figures in high resolution. If you still cannot see the text clearly, please expand the figures up to 500%.

  • It would be better to represent figure 8 in Table, as the comparison between the classes is almost impossible via the figure.

Thank you for the suggestions. We have now presented Figure 8 as Table 2.

  • Please provide a strong conclusion for the manuscript.

Thank you for the advice. The conclusion is now provided in lines 694 to 721.

References:

  1. Seth S, Mallik S, Bhadra T, Zhao Z. Dimensionality reduction and louvain agglomerative hierarchical clustering for cluster-specified frequent biomarker discovery in single-cell sequencing data. Frontiers in Genetics. 2022;13:828479.
  2. Hu Z, Ahmed AA, Yau C. CIDER: an interpretable meta-clustering framework for single-cell RNA-seq data integration and evaluation. Genome Biology. 2021;22:1-21.

Round 2

Reviewer 1 Report

Comments and Suggestions for Authors

Thanks for the revision. 

Reviewer 2 Report

Comments and Suggestions for Authors

I have no further comments.